# Emergency Foot-and-Mouth Disease Vaccines A Malaysia 97 and A_22_ Iraq 64 Offer Good Protection against Heterologous Challenge with A Variant Serotype A ASIA/G-IX/SEA-97 Lineage Virus

**DOI:** 10.3390/vaccines8010080

**Published:** 2020-02-10

**Authors:** Nagendrakumar B. Singanallur, Aldo Dekker, Phaedra L. Eblé, Froukje van Hemert-Kluitenberg, Klaas Weerdmeester, Jacquelyn Horsington, Wilna Vosloo W

**Affiliations:** 1Australian Animal Health Laboratory, CSIRO-Health & Biosecurity, Geelong 3220, Australia; jacquelyn.horsington@merck.com (J.H.);; 2Wageningen Bioveterinary Research (WBVR), Department of Virology, Laboratory Vesicular Diseases, Houtribweg 39, 8221 RA Lelystad, The Netherlands; aldo.dekker@wur.nl (A.D.); phaedra.eble@wur.nl (P.L.E.); kweerd@hotmail.com (K.W.)

**Keywords:** FMD, foot-and-mouth disease virus, vaccine, cross-protection, heterologous protection, vaccine efficacy

## Abstract

The continuous emergence of foot-and-mouth disease virus (FMDV) serotype A variants in South East Asia is of concern for international FMDV antigen banks, especially when in vitro tests predict a low antigenic match. A vaccination-challenge study was performed by using two emergency FMDV vaccines with A_22_ Iraq 64 (A22 IRQ) and A Malaysia 97 (A MAY 97) strains, against challenge with a variant strain of FMDV A/Asia/G-IX/SEA-97 lineage at 7- and 21-day post-vaccination (dpv). At 7 dpv, three of five female calves vaccinated with A MAY 97 and four of five vaccinated with A22 IRQ did not show lesions on the feet and were considered protected, while at 21 dpv all five calves were protected with each vaccine, indicating equal efficacy of both vaccine strains. Calves were protected despite relatively low heterologous neutralizing antibody titers to the challenge virus at the time of challenge. All the calves developed antibodies to the non-structural proteins, most likely due to the direct intradermolingual (IDL) inoculation. Only one calf from the A MAY 97-7 group had infectious virus in the serum 1–3-day post-challenge (dpc), while no virus could be isolated from the serum of cattle challenged on 21 dpv. The virus could be isolated from the oral swabs of all calves, 1–7 dpc with viral RNA detected 1–10 dpc. Nasal swabs were positive for virus 1–6 dpc in a small number of calves. The time between vaccination and infection did not have an impact on the number of animals with persistent infection, with almost all the animals showing viral RNA in their oro-pharyngeal fluid (probang) samples up to 35 dpc. Despite the poor in vitro matching data and field reports of vaccine failures, this study suggests that these vaccine strains should be effective against this new A/Asia/G/SEA-97 variant, provided they are formulated with a high antigen dose.

## 1. Introduction

Foot-and-mouth disease (FMD) is an infectious disease of domestic and wild even-toed animals. The disease can be a major constraint to animal production, especially in dairy cattle, where a reduction in milk yield is often significant, but it also has economic consequences on meat and draught cattle. Due to its infectious nature and potential impact on trade, it is important to prevent the accidental introduction of the virus into a previously “FMD free” country. If an incursion occurs, vaccines can play a vital role in effective control of the disease, both to limit the spread of the virus during epidemics and the economic impact [1]. 

FMD virus (FMDV) exists as seven distinct serotypes (O, A, C, Asia-1, SAT-1, SAT-2 and SAT-3), with numerous topotypes, genotypes or lineages within each serotype [2]. Genetically serotype A viruses have been classified under three major geographically restricted genotypes namely, Euro-SA, Asia and Africa [3] and several genetically distinguishable subgroups [4,5]. In Asia, lineages such as A-Iran87, A-Iran96, A-Iran99 and A-Iran05 have emerged over time and currently a newly emerged genotype, A/Asia/GVII is predominant [6]. However, in Southeast Asia (SEA), since the emergence of the A/ASIA/G-IX/SEA-97 lineage in 1997, there has been little diversity in serotype A viruses, though occasional variants within the lineage have been reported [3,7]. Some of these variant strains have spread beyond SEA, causing outbreaks in countries that were previously free of serotype A, such as the People’s Republic of China and South Korea [8]. However, during 2011–2012, a new variant strain within the SEA-97 lineage emerged in SEA, resulting in vaccine failures in Thailand and Vietnam [9]. Phylogenetic studies, based on nucleotide sequences of the 1D region of serotype A isolates from SEA (2011–2015), showed the emergence of three variant strains during 2004–2008, 2010–2013 and 2014. The OIE Regional Reference Laboratory for FMD in SEA, Pakchong in Thailand, classified these new variants as the A/TAI/LopBuri/2012-related strains. The variant strains showed lower r1 values (poor antigenic match in an in vitro virus neutralization test or ELISA) when matched with the two vaccine strains, A_22_ Iraq 64 (A22 IRQ) and A Malaysia 97 (A MAY 97), which are included in several of (inter)national FMDV antigen banks (unpublished data). 

Current policies to control outbreaks in FMD-free countries include stamping out of all infected animals, as well as movement restrictions and other quarantine measures. In some countries, such as the Netherlands, vaccination is now included in the contingency plan as a standard measure for FMD control [10]. In densely populated livestock areas, the use of emergency vaccines has shown that vaccination has significant advantages in assisting with rapid control of the disease, which is further confirmed in mathematical models [11,12]. In addition, for obvious ethical reasons, and the problem of disposing of large numbers of carcasses, there is a strong desire to reduce the reliance on large-scale culling of animals to control future outbreaks of FMD.

FMDV vaccines are classified either as conventional vaccines with a standard dose of antigen or emergency vaccines with higher doses of antigen. The vaccines incorporate chemically inactivated whole virus preparations of a particular strain or strains, of one or more serotypes, formulated either with an oil-based adjuvant or with aluminum hydroxide-saponin adjuvant [13,14,15]. Vaccine-induced protection against clinical disease is best during homologous challenge, but when antigenically different viruses cause an outbreak, protection can be lower [16]. Several countries that are FMD-free without vaccination have established reserves of inactivated viral antigens in the form of FMDV antigen banks that can be formulated into vaccines when required. These banks hold several FMDV vaccine strains according to the perceived risk for each country or region [17]. Due to the continuous evolution of FMDV, field viruses isolated during outbreaks are generally genetically and antigenically different from vaccine strains. Since it would take several months to produce a homologous vaccine strain during an FMDV emergency, countries with a vaccine bank will select the most suitable strain from the existing antigen reserve. However, it is important to match the vaccine strains as closely as possible to the field strains against which protection is required [18]. In vitro antigen matching studies, using virus neutralization test (VNT) or liquid phase blocking ELISA (LP ELISA), can predict to a certain extent how these vaccine strains match with the field strains. It gives rise to concern if the existing vaccine strains show a poor match (low r_1_-value) with the field virus [17]. However, previous studies have shown that, regardless of a poor antigenic match, most of the high-potency emergency vaccines can protect against heterologous challenge [17,19,20]. In vivo studies using a vaccine potency test such as the protective dose 50 (PD_50_) [17,21,22] or PGP (protection against podal generalization) tests are the only truly reliable way to assess vaccine efficacy [21,23].

In vitro vaccine-matching studies performed by the World Reference Laboratory (WRL), United Kingdom, showed a poor match (r_1_–value < 0.3) with the existing vaccine strains against a lineage of serotype A circulating in SEA, FMDV A/ASIA/G-IX [9]. The objective of the current study was to measure the vaccine efficacy, in cattle, of two vaccine strains (A22 IRQ and A MAY 97) that are present in most FMDV antigen banks, against challenge at 7 and 21 days post vaccination (dpv) to determine the efficiency of vaccines at different time points post-vaccination, as well as their impact on persistent infection.

## 2. Materials and Methods 

### 2.1. Cells and Viruses Used in the Study

Baby hamster kidney (BHK)-21 cells were used for the initial virus isolation of the field isolate in Vietnam (A/VIT/15/2012), as well as for virus neutralization tests (VNT), using A_22_ Iraq 64 (A22 IRQ) and A/VIT/15/2012. IBRS2 cells were used for the VNT, using A Malaysia 97 (A MAY 97). Infectious virus from samples collected during the study was isolated using secondary lamb kidney cells [24]. 

The challenge virus, FMDV isolate A/VIT/15/2012, which belongs to the A/ASIA/G-IX lineage, was obtained from the WRL. It was passaged and titrated in cattle tongue before it was used as the challenge virus and adapted to IBRS2 cells before use in the VNT. Wageningen Bioveterinary Research (WBVR), the Netherlands, supplied A22 IRQ and A MAY 97 viruses adapted to BHK-21 cells for VNT. 

### 2.2. Experimental Animals

Twenty-three female Dutch dairy calves (mainly crossbred Holstein–Friesian, but also some combinations with Fleckvieh, Monbeliarde, Swedish and Norwegian Red and/or red and white Frisian), aged 8–12 months and weighing approximately 200 kg, were used in the high-containment facility at WBVR. All the protocols for experimentation with live cattle were approved by the Animal Ethics Committee of the CSIRO—Australian Animal Health Laboratory (AEC 1679, 1700) and by the Institutional Animal Ethics Committee of WBVR (2014078 LVZ173).

### 2.3. Vaccines

Monovalent double oil emulsion vaccines containing A22 IRQ or A MAY 97 antigen were formulated as emergency vaccines (>6PD_50_/dose) by Merial Animal Health (now Boehringer Ingelheim), United Kingdom. Cattle were vaccinated by administering 2 mL of vaccine, intramuscularly, in the side of the neck, according to the manufacturer’s recommendations.

### 2.4. Preparation of Cattle Challenge Virus

The A/VIT/15/2012 virus was passed once through cattle tongue, using standard procedures [25] and a 10% (w/v) suspension of the lesion material prepared in Minimum Essential Medium with Hanks’ salts with 2% fetal calf serum (FBS) and 2% antibiotics mix (Penicillin 10^5^ IU/100 mL, Streptomycin 0.1g/100 mL, Mycostatin 5 x 10^4^ IU/100 mL, Polymyxin B 15x10^3^ IU/100 mL and Kanamycin 0.1g/100 mL). The titer was 10^7.8^ plaque-forming units (PFU)/mL, as determined on secondary lamb kidney cells and 10^7.3^ cattle infective dose (CID)_50_/mL, when titrated in cattle tongue.

### 2.5. Vaccine Efficacy Studies

Twenty calves were divided randomly and equally into four groups: A22 IRQ-21, A22 IRQ-7, A MAY 97-21 and A MAY 97-7, along with three additional calves that were included as unvaccinated controls (UVC). Groups A22 IRQ-21 and A22 IRQ-7 received 2 mL of monovalent A22 IRQ vaccine, while groups A MAY 97-21 and A MAY 97-7 received 2 mL of monovalent A MAY 97 vaccine. Groups A22 IRQ-21 and A MAY 97-21 were challenged at 21 days post-vaccination (dpv) (D-21 groups), whereas groups A22 IRQ-7 and A MAY 97-7 were challenged at 7 dpv (D-7 groups). Vaccination was staggered to allow the challenge of all calves, including the 3 unvaccinated controls (UVC), on the same day (Table 1).

Calves were challenged by the intra-dermo-lingual (IDL) route with 10^5.2^ PFU/mL of A/VIT/15/2012 at two sites (100 µL per site). The calves were observed daily post-vaccination and post-challenge, until the experiment was terminated at 35 days post-challenge (dpc). Calves were monitored for clinical signs of FMD between 1 and 7 dpc, and detailed examination was carried out on 3 and 7 dpc, following sedation with Xylazine (0.2 mg/kg), which was antagonized by using Atipamezole (0.025 mg/kg). Lesions on the feet were considered signs of systemic spread of infection. Rectal temperatures were monitored daily. Clotted blood was collected on −21, −18, −14, −10, −7 and −4 dpc, daily on 0 to 7 dpc, and 10, 14, 21, 28 and 35 dpc. In the laboratory, blood samples were centrifuged, and serum was collected and stored at −80 °C (for virus isolation and genomic detection) and −20 °C (for serology). Oral and nasal swabs were collected daily from 0 to 7 dpc, and on 10, 14, 21, 28 and 35 dpc, using Salivette swabs (Sarstedt) and sterile cotton swabs, respectively. In the laboratory, oral fluids were extracted using 1 mL of Dulbecco’s minimal essential medium, containing 5% FBS and antibiotics, while the nasal swabs were incubated in 2 mL of Dulbecco’s minimal essential medium, containing 5% FBS and antibiotics. After incubation for 30 min at 4 °C, the tubes were centrifuged at 2000 rpm, and the fluids were collected in new tubes, labeled and stored at −80 °C. Probang samples were collected on −7, 0, 7, 10, 14, 21, 24, 28, 31 and 35 dpc and frozen at −80 °C, until processing.

### 2.6. Serological Assays

Heat-inactivated (56 °C, 30 min) serum samples were used for VNT performed by using standard procedures [26]. Titers were expressed as the final dilution of serum present in the serum/virus mixture, where 50% of wells were protected. Antibodies to the non-structural proteins (NSP) of FMDV were detected using the PrioCHECK® FMDV-NS antibody ELISA (ThermoFisher, Waltham, MA, United States of America), with serum samples diluted at 1:5 and tested in duplicate, following the manufacturer’s instructions. Percentage inhibition (PI) values ≥50 were considered positive.

### 2.7. Virus Isolation and Titration on Cell Culture

Virus isolation and titration were performed, using secondary lamb kidney cells, following standard plaque assay protocols in 6-well collagen plates [27]. The plaques were counted, and results were expressed as log_10_ PFU/mL.

### 2.8. Real-Time RT-PCR Assay for Detection of Viral RNA

Total RNA from serum, nasal swabs and probang samples were isolated using the MagNA Pure 96 DNA and Viral NA Large Volume kit, on the MagNA Pure 96 system (Roche® Life Science). In each run of 96 samples, one negative, one high positive and one moderate positive sample were included as extraction controls. The RT-PCR was carried out as described by the manufacturer (Roche®), using the LightCycler RNA Amplification Kit Hybridisation Probes and LightCycler 480 (Roche® Life Science), using the protocol described by Moonen et al. [28]. In each run of 96 samples, one negative, one high positive and one moderate positive RT-PCR control were included. The test protocol was as follows: reverse transcription for 20 min at 61 °C, denaturation for 1 min at 95 °C, followed by 45 PCR cycles of 3 s at 95 °C, 15 s at 60 °C and 10 s at 72 °C. Amplification was monitored in real time, using hybridization probes. Samples were considered positive when the fluorescence signal rose above the background signal (crossing point determined automatically by the second derivative maximum method for quantification by the software supplied by Roche®) [28].

### 2.9. Statistical Analyses

Clinical protection (count data) were analyzed using the Fischer exact test. For analysis of the VNT data, ANOVA (one-way for a single factor or two-way for multiple factors) was used to identify significant differences between groups. If a statistical difference was identified by using the one-way ANOVA, then the result was analyzed in a pairwise t-test (using Holm correction for multiple comparisons). In the two-way ANOVA, significant differences between models were tested by using the F-test. Longitudinal data (virus isolation, RT-PCR results and NSP response) were analyzed, using a linear mixed regression model, using the LME4 Library [29], with animal number as random variable and dpc, group and vaccination (yes or no) as possible explanatory variables. Using forward selection, the best model with the lowest AIC (Akaikes Information Criterion) was chosen. For the NSP responses, the data from 0 to 35 dpc were analyzed. NSP-response (PI) was used as a response variable. Calf number was added as a random variable. As explanatory variables, dpc (as a factor) and vaccine group were analyzed, as well as interactions. Data on virus isolation and RT-PCR were also analyzed the same way. All analyses were performed by using R version 3.3.1 [30]. All the results are compiled as Appendix A.

## 3. Results

### 3.1. Clinical Outcome Post-Challenge

Pyrexia (body temperature >40 °C) that lasted 1–4 dpc was observed in all vaccine groups and unvaccinated controls post-challenge (results not shown). One of the calves in the UVC group was humanely euthanized for ethical reasons, 11 days after challenge, due to rumen atony and dysfunction that did not resolve.

All calves in the D-21 groups (A22 IRQ-21 and A MAY97-21) were clinically protected against FMD (i.e., no lesions observed on the feet). One calf in the A22 IRQ-7 (#8105) group had FMD lesions on the right front foot, while two of the calves in A MAY97-7 had lesions on three (#8109) or four feet (#8111). All three UVC calves (#8112, #8113 and #8114) showed generalized disease, with foot lesions on all four feet (Table 1). No significant difference in protection was observed between the A22 IRQ-21 and AMAY97-21 vaccine groups and between the A22 IRQ-7 and AMAY97-7 vaccine groups (Fischer exact test).

### 3.2. Neutralizing Serological Response 

Neutralizing antibodies determined by using the VNT with the same virus as the vaccine strain were considered as homologous, while antibodies in the VNT against the challenge virus were considered heterologous. In the statistical analysis, for titers with a value of <0.3 (log_10_), 0.15 was used as a value and for titers with a value of 2.4 (log_10_), and 2.55 was used as a value. A fourfold increase in homologous neutralizing antibody titers was observed at 7 dpv in all four vaccine groups (Figure 1A–C). The mean homologous neutralizing antibody titers on the day of challenge was 2.2 log_10_ (SD = 0.20) for the A22 IRQ-21 and 2.0 log_10_ (SD = 0.20) for the A MAY97-21 group, compared to 1.5 log_10_ (SD = 0.30) and 1.6 log_10_ (SD = 0.23) for A22 IRQ-7 and A MAY97-7 groups, respectively (Figure 1A,B; Figure 2A,B; Table 2). There was a significant difference in homologous antibody titers between the A22 IRQ groups (A22 IRQ-21 vs. A22IRQ-7; t-test; *p* = 0.002) and between the A MAY 97 groups (A MAY97-21 vs. A MAY97-7; t-test; *p* = 0.01). 

Heterologous neutralizing antibody titers against A/VIT/15/2012 (≥1.20 log_10_) were observed at 10 dpv in 9 out of 10 calves in the D-21 vaccine groups (except for #8101 in A MAY97-21) (Figure 1C; Table 1). The mean heterologous antibody titer in both the A22 IRQ-21, as well as the A MAY97-21 group, reached its peak at 10–14 days dpv and were 1.35 log_10_ (SD = 0.28) and 1.11 log_10_ (SD = 0.25), respectively, and in the D-7 vaccine groups, it was 1.05 log_10_ (SD = 0.18) and 0.69 log_10_ (SD = 0.35) for A IRQ-7 and A MAY97-7 groups, respectively, at the time of challenge. There was no significant difference in heterologous antibody titers for the two A22 IRQ vaccinated groups (A22 IRQ-21 vs. A22IRQ-7, *p* = 0.094) or for the two A MAY 97 vaccinated groups (A MAY97-21 vs. A MAY97-7, *p* = 0.06). No differences were found between the groups that were vaccinated at -21 dpc (A22 IRQ-21 vs. A MAY97-21; 0.019), or between the groups that were vaccinated at -7 dpc (A22 IRQ-7 vs. A MAY97-7; *p* = 0.07). In both the A22 IRQ-7 and the A MAY97-7 groups, titers increased after challenge, but it is unclear if this was due to vaccination or challenge. Post-challenge all the vaccine groups showed a more than fourfold increase in neutralizing antibody titer against the challenge virus. 

### 3.3. NSP-Responses

Two calves in the A22 IRQ-21 group (#8095 and #8096) and one in the A22 IRQ-7 group (#8105) showed low positive results in the NSP ELISA on at least one day before challenge (PI values 50%–52%). Most probably, these were non-specific reactions, because, on other days, the PI values were below 50%. At 6–7 dpc, all animals from the vaccine groups had NSP antibody responses that lasted until 35 dpc, when the experiment was terminated (Figure 3). Amongst the UVC group, calf #8113 showed a low positive response from 0 to 6 dpc (52%–57% inhibition), with a strong increase above 80% inhibition from 8 dpc. Calf #8112 in the UVC group had a strong response (> 90% inhibition) between 10 and 21 dpc, and the percentage inhibition declined on day 28 and 35 dpc. The linear mixed regression model analysis showed that dpc (as a factor), group and interaction between dpc and group best explained the NSP response (Appendix A). The interaction shows that there was a significant difference between the different groups in the peak of the response at different days post challenge, which is mainly due to the lower response in the unvaccinated controls after 21 dpc (Figure 3).

### 3.4. Detection of Infectious Virus and Viral RNA in Various Samples Post-Challenge

#### 3.4.1. Serum Samples

In total, 19 of the 20 vaccinated calves had no infectious virus in the serum, and viral RNA could be detected in ten of the calves at 1 dpc; a number had viral RNA intermittently; and all calves were negative for FMDV genome from 5 dpc onward (Appendix A). One protected calf had three positive RT-PCR results from 1–4 dpc, whilst an unprotected vaccinated animal had infectious virus and viral RNA detected until 4 dpc (#8111), (Table 3). In contrast, viral RNA was detected in the sera of all three unvaccinated controls from 1–5 dpc and virus was isolated 1-3 dpc (Table 3). There were more than twice as many positive RT-PCR results (*n* = 26) in blood, compared to VI positive results (*n* = 12), possibly as a result of the latter being more sensitive to presence of neutralizing antibodies. The serum samples did not show any significant difference in infectious virus levels between the different vaccine groups or days after infection; however, significantly higher viral RNA levels were found in the UVC group, but this varied by day after challenge (Appendix A).

#### 3.4.2. Oral and Nasal Fluids 

Infectious virus and viral RNA were detected in the oral fluids of all calves (Table 3 and Appendix A). Viral RNA was detected for longer periods compared to infectious virus. Both infectious virus and viral RNA were detected more frequently and at higher levels in oral swabs (*n* = 179) than in nose swabs (*n* = 131). Viral RNA could be detected in the oral fluid at 14 dpc from one calf in the A22 IRQ-7 group (#8106) and two calves in the A MAY 97-7 group (#8109 and #8111), while viral RNA was detected again at 28 dpc in #8109. In the linear mixed regression model, the virus titers in oral swabs showed a significant difference between days post-infection but did not show any significant difference in infectious virus levels between the different vaccine groups. The same analysis for virus titers in nose swabs samples did not show significant differences between groups or time post-challenge. However, in oral swabs, as well as nose swabs, the linear mixed regression model showed a significant difference in the RNA levels between days after infection and group, with significant differences between groups on different days (interaction between days after infection and group was significant) (Appendix A).

#### 3.4.3. Probang Samples

FMDV was isolated from probang samples from 15 of the 23 infected calves, whilst viral RNA was detected in all calves in at least one sample in the period between 10 and 35 dpc (Table 4). By 35 dpc, seven calves in the D-21 group and nine in the D-7 group had viral RNA in the probang samples, but no infectious virus could be detected in these samples (Appendix A). There was no significant difference in the number of positive calves between groups (Fischer exact test: *p* = 0.90). Both the control animals had viral RNA in the probang samples, but infectious virus could not be isolated. The titers in probang samples did not exceed 2.18 PFU/mL, in contrast to the highest titers in oral (7.19 PFU.mL) and nasal swabs (3.88 PFU/mL). The linear mixed regression model did not show any significant difference in infectious virus levels in probangs between the different groups or days after infection; however, a significant difference in the RNA levels was noticed between days after infection and group and between groups at different days (interaction between day after infection and group was significant; Appendix A).

## 4. Discussion

The poor in vitro vaccine matching data, combined with field evidence of vaccine failures in SEA (WRL FMD Report 2013), suggested that the FMDV vaccine strains, A MAY 97 and A22 IRQ, might be ineffective against the new variants of A/ASIA/G-IX/SEA-97 lineage. The study described here measured the efficacy of the abovementioned vaccine strains against challenge with an FMDV variant virus of A/Asia/G-IX/SEA-97 lineage at 7 dpv to determine early protection and at 21 dpv when the immune response is fully developed. The vaccines conferred partial protection in 60%–80% of the calves by 7 dpv, and 100% protection in those challenged on 21 dpv. The time interval between the vaccinations and infection is unpredictable in an outbreak, and this study confirms previous reports that the number of days between vaccination and infection significantly influences the outcome whilst emergency vaccines with a higher antigen content can be effective despite a poor antigenic match [17,22,31,32,33,34]. It has also been suggested that vaccination reduces transmission (reproduction ratio below 1; R_0_ < 1), even if animals become infected, mainly by reduction of infectivity, which is probably correlated with the reduction of clinical disease and excretion [35]. Several studies showed that reduction of transmission can be achieved within 14 days after vaccination [35,36,37,38,39]. There are strong indications that virus excretion is reduced when a challenge occurs even earlier post-vaccination [34,40,41]. Therefore, in case of an FMD incursion, an early decision for emergency vaccination is necessary to allow enough opportunity for a mature immune response and better protection [32,34].

All calves in the D7 and D21 vaccine groups had detectable homologous and heterologous neutralizing antibody titers on the day of the challenge, indicating a successful outcome to the emergency vaccination, even though the mean antibody titers between the D-21 and D-7 groups differed significantly. The heterologous neutralizing antibody titers to the challenge virus were lower than the homologous titers in most of the animals, but apparently enough to offer clinical protection in all but three calves, despite the severe direct challenge by IDL route. Under field conditions, it is likely that exposure to FMDV will be less severe and therefore it can be expected that the clinical protection would be better than what was observed in this study [34]. During a response to an outbreak, when movement restrictions will be enforced, exposure will be further mitigated. In such a situation, good protection from the clinical disease against challenge at 7 dpv can be expected. Similar observations were made by Cox et al. [42]. However, the mechanisms that govern the early protection in the absence of neutralizing antibodies are poorly studied and understood. The role of the innate immune response along with the non-neutralizing and opsonizing antibodies (that is detectable by ELISA) and IgM needs further investigation. The in vitro correlates responsible for early protection offered by emergency vaccines should be identified to provide confidence to the different stakeholders during vaccine assisted control of FMD outbreaks in ‘*FMD free*’ countries.

Viremia, defined by isolating infectious FMDV, was not found in any of the challenged vaccinated calves, except for one in the A MAY 97-7 group. This contrasted with the UVC group that showed viremia 1–3 dpc. The linear mixed regression model confirmed differences between the level of viral RNA detected in vaccinated and control groups in blood, mouth swabs, nose swabs and probangs. Protection from clinical disease did not correlate with the prevention of sub-clinical infection, as measured by the presence of anti-NSP antibodies, irrespective of clinical status. This confirms the findings of Cox et al. [42] who investigated cattle vaccinated 10 days prior to exposure to other infected animals. However, in earlier studies, it was shown that very few animals seroconverted to NSP when cattle were vaccinated 21 days prior to challenge [43,44] and challenged by direct contact with diseased animals where there was a good match between the vaccine and the challenge virus (r1 value >0.30). This contrasts with our study where the calves were challenged by IDL route and there was a poor antigenic match between the vaccines and the challenge virus.

Since the calves were challenged by IDL route, the primary site of replication was the tongue epithelium with lesions developing on the tongue. FMDV could be isolated from the oral swabs of all calves between 1 and 7 dpc, with viral RNA detected by PCR between 1and 10 dpc. One animal in the A MAY 97-7 and two in A22 IRQ-7 groups had viral RNA up to 14 dpc. The result suggests that the D21 groups had limited viral replication at the primary site of infection compared to D7 groups, showing an advantage of a more mature immune response and therefore early vaccination. It also provided some evidence of the value of oral swabs as diagnostic material where tongue lesions are predominant with a longer diagnostic window, although our observation might be biased by the IDL challenge method. FMDV at low titers was isolated from nose swabs, only from a few vaccinated calves between 1 and 6 dpc.

Earlier studies have shown that the percentage of animals that remain persistently infected after an outbreak of FMD is up to 50% in unvaccinated herds, decreasing considerably with vaccination, as vaccination reduces transmission [45]. Reducing the number of persistently infected animals in a herd will have a much bigger impact on the risk of disease maintenance and transmission [11,12], though the risk associated with such animals is relatively low [46]. In our study, the time between vaccination and infection did not have an impact on the number of animals becoming persistently infected. Almost all the animals had viral RNA in their probang samples at least until 35 dpc, and the levels of RNA did not differ significantly between the groups. Infectious virus at very low titers could be isolated intermittently until 31 dpc from at least one or two animals in each group. Similar observations were made by Cox and Barnett [32], where no differences in virus recovery rates from OPF were apparent between vaccine treatment groups or in comparison to the unvaccinated treatment groups. However, in the study mentioned before, where cattle were challenged 21 dpv by direct contact with an antigenically related virus, virus/viral RNA was recovered less frequently compared to unvaccinated groups [44]. It is probable that these factors play a role in the number of persistently infected animals and should be considered during an outbreak response. 

## 5. Conclusions

The vaccines used in this study are widely used in Southeast Asia (A MAY 97) and in the Middle East (A22 IRQ). Despite the poor in vitro matching data and field reports of vaccine failures, this study suggests that these vaccine strains should be effective against this new A/Asia/G-IX/SEA-97 variant, provided they are formulated with a high antigen dose. The findings further support the hypothesis that highly potent emergency vaccines can provide protection against heterologous challenge, when used correctly, and offer early protection, leading to a reduction in virus excretion. However, in FMD free countries, vaccination should always be one option that needs to be combined with other measures, such as movement control and stamping out, to ensure control and subsequent eradication of FMD.

## Figures and Tables

**Figure 1 vaccines-08-00080-f001:**
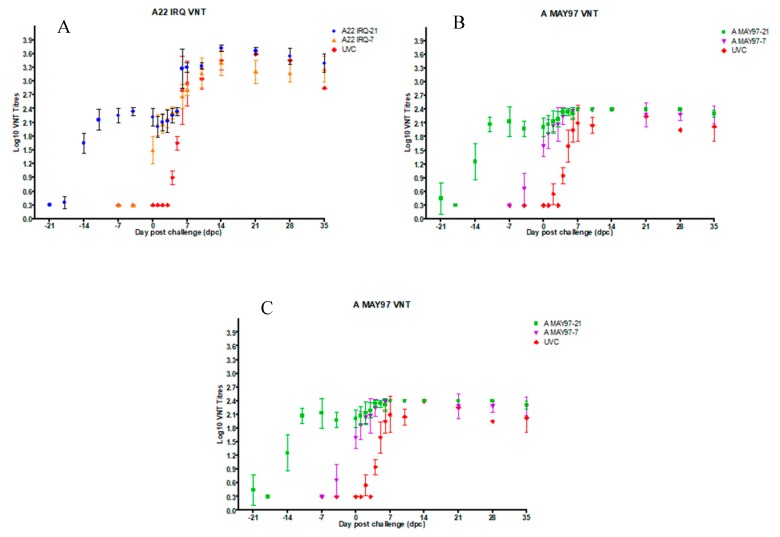
Mean antibody titers determined by VNT (expressed as log_10_ virus neutralization titers) in calves vaccinated with A22 IRQ and A MAY 97 monovalent vaccine and UVC group. Viruses used in the VNT were A22 IRQ (**Panel A**), A MAY 97 (**Panel B**) and A/VIT/15/2012 (**Panel C**). The calves were challenged on 21 dpv (A22 IRQ-21, A MAY 97-21) or 7 dpv (A22 IRQ-7, A MAY97-7). UVC group was challenged on the same day. The error bars indicate the standard error of the mean antibody titers for each group. For the A MAY 97 and A/VIT/15/2012 VNT, titers >2.4 are expressed as 2.4.

**Figure 2 vaccines-08-00080-f002:**
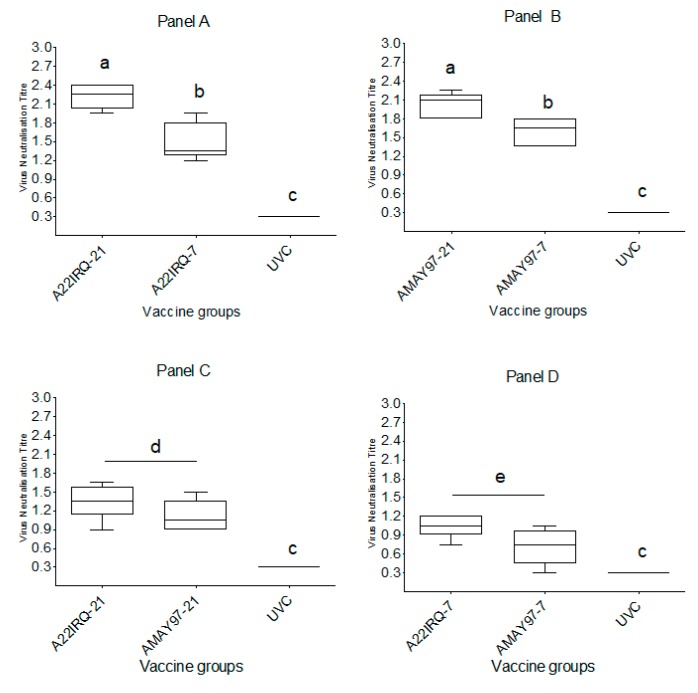
Serum antibody response (log_10_) in calves vaccinated with A22 IRQ and A MAY 97 monovalent vaccines and UVC calves estimated by VNT, using A22 IRQ (**Panel A**), A MAY 97 (**Panel B**) and A/VIT/15/2012 (**Panel C, D**) on the day of challenge. Statistical differences between the groups were measured using a one-way ANOVA test; within each graph, groups with same superscripts do no differ significantly, *p* < 0.05).

**Figure 3 vaccines-08-00080-f003:**
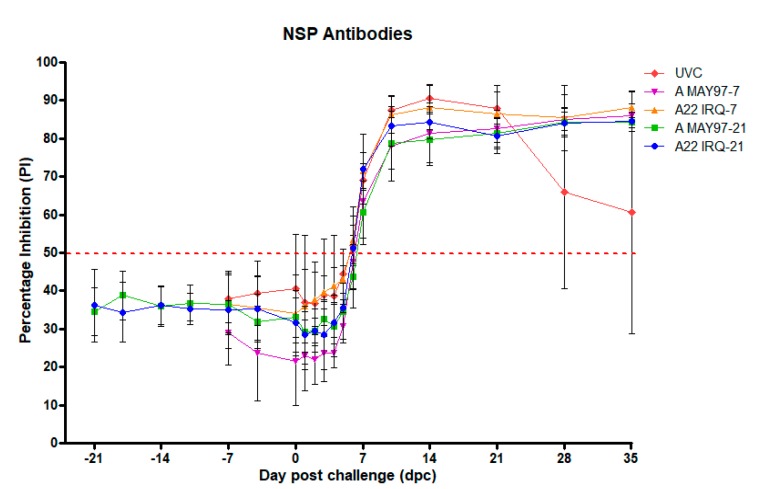
The mean response to the non-structural proteins (NSP) measured as percentage inhibition (PI values), using the FMD NS ELISA kit. Calves were vaccinated with A22 IRQ and A MAY 97 monovalent vaccines, or left unvaccinated (UVC) and challenged with A VIT/15/2012. The error bars indicate the standard deviation for each group. The horizontal dotted red line indicates the cutoff (50% PI).

**Table 1 vaccines-08-00080-t001:** Animal groups, vaccination and challenge.

Group	Vaccination	Challenge day post-vaccination with infectious FMDV A/VIT/15/2012
A22 IRQ-21	Calves, *n* = 5, Vaccinated with 2 mL A22 Iraq 64 Monovalent oil Adjuvanted Vaccine	21
A MAY 97-21	Calves, *n* = 5, Vaccinated with 2 mL A Malaysia 97 monovalent oil adjuvanted vaccine	21
A22 IRQ-7	Calves, *n* = 5, Vaccinated with 2 mL A22 Iraq 64 monovalent oil adjuvanted vaccine	7
A MAY 97-7	Calves, *n* = 5, Vaccinated with 2 mL A Malaysia 97 monovalent oil adjuvanted vaccine	7
UVC	Calves, *n* = 5, Unvaccinated controls	On the same day as the vaccine groups

**Table 2 vaccines-08-00080-t002:** Homologous and heterologous post-vaccination neutralizing antibody titers in calves vaccinated with either A22 IRQ or A MAY 97 monovalent vaccines and UVC group, expressed as log_10_ values at the time of challenge. All sera were tested against A22 IRQ, A MAY 97 and A/VIT/15/210. Titers ≥1.20 log_10_ are considered positive. FMD Lesions: RF = right forelimb; LF = left forelimb; BF = both forelimbs; BH = both hindlimbs.

Group	Animal ID	A22 IRQ	A MAY 97	A/VIT/15/2012	FMD Lesions
A22 IRQ-21	8092	2.25	1.20	1.65	No
	8093	2.40	1.35	1.50	No
	8094	2.10	0.60	0.90	No
	8095	1.95	1.05	1.35	No
	8096	2.40	1.35	1.35	No
A MAY 97-21	8097	0.60	1.80	0.90	No
	8098	1.05	2.10	1.20	No
	8099	0.90	2.25	1.05	No
	8100	1.20	2.10	1.50	No
	8101	0.75	1.80	0.90	No
A22 IRQ-7	8102	1.95	0.60	1.20	No
	8103	1.35	<0.30	1.05	No
	8104	1.65	0.60	1.20	No
	8105	1.35	0.90	1.05	RF
	8106	1.20	0.60	0.75	No
A MAY 97-7	8107	0.60	1.80	0.90	No
	8108	0.75	1.65	1.05	No
	8109	<0.30	1.35	<0.30	LF, BH
	8110	<0.30	1.80	0.75	No
	8111	<0.30	1.35	0.60	BF, BH
UVC	8112	<0.30	<0.30	<0.30	BF, BH
	8113	<0.30	<0.30	<0.30	BF, BH
	8114	<0.30	<0.30	<0.30	BF, BH

**Table 3 vaccines-08-00080-t003:** Viremic phase: virus isolation indicated as log_10_ PFU per mL of FMDV A/VIT/15/2012 and viral RNA detected in sera and swabs from 0 to 35 days post-challenge. Animals were vaccinated with two different vaccines and challenged at 7- or 21-days post-vaccination, with a heterologous virus, along with UVC group.

Group		0 dpc				1 dpc			2 dpc			3 dpc			4 dpc			5 dpc			6 dpc			7 dpc			
	Animal #	S	OS	NS	P	S	OS	NS	S	OS	NS	S	OS	NS	S	OS	NS	S	OS	NS	S	OS	NS	S	OS	NS	P
**A22 IRQ-21**	8092	-	-	-	-	-	5.01	0.40	-	4.24	-	-	5.32	0.40	-	2.57	-	-	4.13	-	-	-	-	-	-	-	-
	8093	-	-	-	-	-	1.00	0.40	-	2.46	-	-	4.25	-	-	3.24	-	-	4.12	-	-	4.25	-	-	2.18	-	-
	8094	-	-	-	-	-	6.31	0.70	-	4.85	2.48	-	5.30	2.59	-	4.28	1.89	-	3.49	1.10	-	2.00	-	-	0.40	-	-
	8095	-	-	-	-	-	5.39	-	-	4.89	-	-	4.27	0.70	-	4.13	-	-	3.36	-	-	1.90	-	-	0.00	-	-
	8096	-	-	-	-	-	3.98	-	-	3.86	-	-	3.79	-	-	4.43	-	-	5.09	-	-	2.60	-	-	0.00	-	-
**A MAY 97-21**	8097	-	-	-	-	-	6.91	-	-	5.48	1.70	-	4.34	-	-	4.01	-	-	3.00	-	-	1.74	-	-	-	-	-
	8098	-	-	-	-	-	5.89	-	-	7.37	-	-	4.20	-	-	3.92	0.40	-	5.05	-	-	2.04	-	-	0.70	-	-
	8099	-	-	-	-	-	3.08	-	-	5.00	-	-	3.40	0.70	-	3.18	-	-	1.40	-	-	-	-	-	-	-	0.40
	8100	-	-	-	-	-	1.10	-	-	3.56	-	-	2.40	-	-	3.25	-	-	1.57	-	-	-	-	-	-	-	0.40
	8101	-	-	-	-	-	6.07	2.72	-	4.47	-	-	3.82	-	-	3.13	1.30	-	1.74	-	-	2.15	-	-	1.24	-	0.70
**A22 IRQ-7**	8102	-	-	-	-	-	5.88	-	-	3.47	-	-	3.25	0.40	-	3.10	-	-	5.44	-	-	3.30	-	-	2.41	-	-
	8103	-	-	-	-	-	5.45	-	-	4.30	1.18	-	2.88	1.18	-	3.40	1.93	-	3.47	-	-	3.10	1.65	-	-	-	0.88
	8104	-	-	-	-	-	5.82	-	-	2.72	-	-	2.72	-	-	2.70	-	-	1.76	-	-	0.40	-	-	-	-	-
	8105*	-	-	-	-	-	5.70	-	-	3.86	-	-	3.36	-	-	5.76	-	-	3.00	-	-	1.97	-	-	0.40	-	1.30
	8106	-	-	-	-	-	5.56	-	-	5.68	-	-	3.72	-	-	4.31	-	-	4.40	-	-	2.70	-	-	2.70	-	-
**A MAY 97-7**	8107	-	-	-	-	-	4.05	-	-	4.11	-	-	2.53	-	-	2.38	-	-	2.88	-	-	1.44	-	-	-	-	-
	8108	-	-	-	-	-	3.99	-	-	5.06	-	-	3.10	-	-	3.00	-	-	2.00	-	-	1.48	-	-	-	-	1.81
	8109*	-	-	-	-	-	6.15	0.40	-	3.60	1.80	-	2.70	-	-	1.30	0.40	-	1.80	-	-	0.88	-	-	-	-	-
	8110	-	-	-	-	-	5.63	-	-	6.01	-	-	3.70	-	-	5.11	0.70	-	3.13	-	-	2.88	-	-	-	-	-
	8111*	-	-	-	-	3.25	4.00	3.14	3.27	4.77	3.18	2.27	4.33	3.16	-	3.70	2.62	-	2.88	1.10	-	0.70	-	-	0.70	-	-
**Unvaccinated Controls**	8112*	-	-	-	-	3.52	2.59	-	3.16	4.46	2.77	3.20	4.89	2.60	-	4.52	3.02	-	1.80	1.74	-	0.40	-	-	1.30	-	-
	8113*	-	-	-	-	3.17	5.98	-	2.74	4.79	1.94	2.97	3.10	-	-	3.74	-	-	4.72	-	-	4.44	-	-	2.50	-	2.18
	8114*	-	-	-	-	3.42	7.19	0.70	3.30	5.42	3.00	3.12	4.15	2.80	-	3.63	3.88	-	2.70	1.44	-	3.72	1.54	-	4.46	-	-

Notes: dpc = days post challenge; S = serum, OS = oral swab, NS = nasal swab, P = probang sample; cells with gray shade are positive for FMDV genome; “-” indicates below the limit of detection by PCR or VI;. * - Calves that were not protected against challenge.

**Table 4 vaccines-08-00080-t004:** Post-viremic phase: virus isolation indicated as log_10_ PFU per mL of FMDV A/VIT/15/2012 and viral RNA detected in sera and swabs from 0 to 35 days post-challenge. Animals were vaccinated with two different vaccines and challenged at 7- or 21-days post-vaccination, with a heterologous virus, along with UVC group.

Groups		10 dpc				14 dpc				21 dpc				28 dpc				30 dpc				35 dpc			
	Animal #	S	OS	NS	P	S	OS	NS	P	S	OS	NS	P	S	OS	NS	P	S	OS	NS	P	S	OS	NS	P
**A22 IRQ-21**	8092	-	-	-	0.70	-	-	-	-	-	-	-	-	-	-	-	-	X	X	X	-	-	-	-	-
	8093	-	-	-	-	-	-	-	-	-	-	-	-	-	-	-	-	X	X	X	-	-	-	-	-
	8094	-	-	-	-	-	-	-	1.24	-	-	-	2.09	-	-	-	1.00	X	X	X	-	-	-	-	-
	8095	-	-	-	-	-	-	-	-	-	-	-	-	-	-	-	-	X	X	X	-	-	-	-	-
	8096	-	-	-	-	-	-	-	-	-	-	-	-	-	-	-	-	X	X	X	-	-	-	-	-
**A MAY 97-21**	8097	-	-	-	-	-	-	-	-	-	-	-	-	-	-	-	-	X	X	X	-	-	-	-	-
	8098	-	-	-	-	-	-	-	-	-	-	-	-	-	-	-	-	X	X	X	1.54	-	-	-	-
	8099	-	-	-	-	-	-	-	-	-	-	-	-	-	-	-	-	X	X	X	-	-	-	-	-
	8100	-	-	-	-	-	-	-	-	-	-	-	-	-	-	-	-	X	X	X	1.35	-	-	-	-
	8101	-	-	-	-	-	-	-	-	-	-	-	-	-	-	-	-	X	X	X	-	-	-	-	-
**A22 IRQ-7**	8102	-	-	-	-	-	-	-	-	-	-	-	-	-	-	-	-	X	X	X	-	-	-	-	-
	8103	-	-	-	-	-	-	-	1.54	-	-	-	-	-	-	-	-	X	X	X	-	X	X	X	X
	8104	-	-	-	-	-	-	-	1.35	-	-	-	-	-	-	-	-	X	X	X	-	-	-	-	-
	8105*	-	-	-	-	-	-	-	-	-	-	-	-	-	-	-	-	X	X	X	-	-	-	-	-
	8106	-	-	-	-	-	-	-	1.30	-	-	-	1.18	-	-	-	-	X	X	X	-	-	-	-	-
**A MAY 97-7**	8107	-	-	-	-	-	-	-	-	-	-	-	-	-	-	-	-	X	X	X	1.72	-	-	-	-
	8108	-	-	-	-	-	-	-	1.85	-	-	-	-	-	-	-	-	X	X	X	-	-	-	-	-
	8109*	-	-	-	-	-	-	-	-	-	-	-	-	-	-	-	-	X	X	X	-	-	-	-	-
	8110	-	-	-	-	-	-	-	-	-	-	-	-	-	-	-	-	X	X	X	-	-	-	-	-
	8111*	-	-	-	1.24	-	-	-	1.24	-	-	-	1.00	-	-	-	1.30	X	X	X	-	-	-	-	-
**Unvaccinated Controls**	8112*	-	-	-	-	-	-	-	-	-	-	-	-	-	-	-	-	X	X	X	-	-	-	-	-
	8113*	-	-	-	-	-	-	-	-	-	-	-	-	-	-	-	-	X	X	X	-	-	-	-	-
	8114*	-	-	-	1.00	Animal Euthanized																			

Notes: dpc = day post challenge; S = serum, OS = oral swab, NS = nasal swab, P = probang sample; cells with gray shade are positive for FMDV genome; “-” indicates below the limit of detection by PCR or VI; X = no sampling done on that day. * - Calves that were not protected against challenge.

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
