# Peer review of "Emergency Foot-and-Mouth Disease Vaccines A Malaysia 97 and A22 Iraq 64 Offer Good Protection against Heterologous Challenge with A Variant Serotype A ASIA/G-IX/SEA-97 Lineage Virus"

_vaccines, 2020, doi:10.3390/vaccines8010080_

Round 1

Reviewer 1 Report

The manuscript “Emergency foot-and-mouth disease vaccines A MAY 97 and A22 IRQ 64 offer good protection against heterologous challenge with a variant serotype A/ASIA/G-IX/SEA-97 lineage virus” presents the results of a vaccination-challenge study on cattle by using two vaccines A22 IRQ and A MAY 97, commonly present in international vaccine banks, and as challenge virus a variant strain circulating in South East Asia (SEA) belonging to A/Asia/GIX/SEA-97 lineage.

The topic of the manuscript is of high interest, during a foot-and-mouth disease outbreak, very often in developing countries, only heterologous vaccines are available, and it is urgent knowing the performances of heterologous vaccines.

The manuscript has been well organized and the study has been well carried out and conceived. The results are consistent with the discussion and the conclusion.

For all the explained reasons, it is worth publishing this manuscript.

Minor spelling mistakes and text editing:

Line 93: please check how is written “In vitro”

Line 103: No information has been given about A/IRN/22/2018 strain

Line223-224: “22” is missing in the name of the vaccine strain A22IRQ-7

Line 263: Please check the number of the tables which authors are referring to in the paragraph

Author Response

Minor spelling mistakes and text editing:

Line 93: please check how is written “In vitro”

Corrected

Line 103: No information has been given about A/IRN/22/2018 strain

Corrected to A/VIT/15/2012

Line223-224: “22” is missing in the name of the vaccine strain A22IRQ-7

To be consistent with the vaccine strains and groups the following changes have been made throughout the manuscript: Vaccine strains: A22 Iraq 64 (A22 IRQ) and A Malaysia 97 (A MAY 97); Vaccine groups: A22 IRQ-21, A22 IRQ-7, A MAY97-21 and A MAY97-7 and unvaccinated and challenged control group: UVC.

Line 263: Please check the number of the tables which authors are referring to in the paragraph

Corrected as Table 3a & 3b due to addition of a new Table 1.

Reviewer 2 Report

This study addressed the effectiveness of banked FMDV vaccines against a newly emerged isolate.  This is a large animal study conducted in high containment and it's not very often these types of needed studies are performed.

I had a few concerns with the manuscript.

The abstract needs to be more clear.  The first 3 sentences are incredible confusing.  "serotype A" = of what? define FMDV.  "emergency vaccines" - of what?  methods 2.5 please add text regarding your control animals.  Surely you had an infected, non-vaccinated and a non-infected, non-vaccinated. Results. 3.1.1 was the euthanized control cow infected, non-vaccinated or non-infected, non-vaccinated? Results. 3.1.3. Are these NSP positive control cows infected, non-vaccinated or non-infected, non-vaccinated.  please clarify.  If it is the later, please add discussion on why this as observed. Results 3.1.4.3. Again with the control animals...please clarify Please indicate in your tables whether you unvaccinated controls were infected. Discussion: again, please clarify the infection status of your control animals.

Author Response

The abstract needs to be more clear.  The first 3 sentences are incredible confusing.  "serotype A" = of what? define FMDV.  "emergency vaccines" - of what? 

Changed as follows: The continuous emergence of foot-and-mouth disease virus (FMDV) serotype A variants in South East Asia is of concern for international FMDV antigen banks, especially when in vitro tests predict a low antigenic match.  A vaccination-challenge study was performed using two emergency FMDV vaccines with A22 Iraq 64 (A22 IRQ) and A Malaysia 97 (A MAY 97) strains, against challenge with a variant strain of FMDV A/Asia/G-IX/SEA-97 lineage at 7- and 21- day post vaccination (dpv). 

Methods 2.5 please add text regarding your control animals.  Surely you had an infected, non-vaccinated and a non-infected, non-vaccinated.

Results. 3.1.1 was the euthanized control cow infected, non-vaccinated or non-infected, non-vaccinated?

Results. 3.1.3. Are these NSP positive control cows infected, non-vaccinated or non-infected, non-vaccinated.  please clarify.  If it is the later, please add discussion on why this was observed

Results 3.1.4.3. Again with the control animals...please clarify

Please indicate in your tables whether you unvaccinated controls were infected.

Discussion: again, please clarify the infection status of your control animals. 

Response: In section 2.5 we have included a new table (Table 1) as suggested by Reviewer 3 to describe the different groups.  The four vaccine groups have been identified as A22 IRQ-21, A MAY97-21, A22 IRQ-7 and A MAY97-7 and the unvaccinated control calves as UVC.  All calves were challenged on the same day.  As a result, in the subsequent sections (Results 3.1.1, 3.1.3, 3.1.3.3, Tables 1, 2 and 3, and Discussion) we have referred the unvaccinated and challenged calves as UVC group.

Since the work was performed in a high containment facility, all the animals must be destroyed and cannot be used for any other studies.  Therefore, for animal ethics reasons (reduction) we are not allowed to have unvaccinated and uninfected animals in the study.  Also, they do not serve any purpose for statistical comparison of results.

Reviewer 3 Report

In this manuscript Singanallur and colleagues describe their studies of efficacy of “emergency vaccination” against foot and mouth disease virus (FMDV) using heterologous virus strains with low in vitro antigenic match to the challenge virus. The authors employ a challenge model with intradermolingual inoculation of virus, which probably is a more severe challenge model than the natural infection route, but ensures that all animals receive the same amount of virus inoculum. They define clinical disease as lesions on the feet, while ignoring/disregarding any oral lesions. This seems odd since virus is more likely to be spread via saliva and nasal secretions and oropharyngeal fluid (probang samples) are used to assess viral persistence. Challenge with FMDV was done 7 and 21 days post vaccination to simulate likely time frames during an epidemic, where vaccination rather than culling is a chosen manner of containment also combined with restrictions on animal movements. They saw protection against foot-lesions in all vaccinated animals except for two animals challenged 7 days post vaccination. However, virus was isolated from probang samples up to 30 days post challenge and viral RNA detected in all but one animal at 35 days post challenge (the last sampling day), indicating that vaccination does not protect against virus persistence and subclinical infection – as is also the experience from natural settings (e.g., DOI: 10.1111/tbed.12774; DOI: 10.1111/tbed.12963). Hence, even if movement restrictions are implemented at least until no further clinical signs are evident, there is no guarantee that the virus has been eradicated. While this may be of greater concern in endemic areas where probang testing of every animal might be beyond both practical and economic capabilities, it is worth taking into consideration for non-endemic areas, if FMDV is introduced and vaccination combined with movement restriction is considered an option. In high-density areas like Europe and parts of North-America, there would also have to be long-term control of fomites, trade etc as well as extensive testing. And in a country like Australia it would be next to impossible to test all cattle, considering the management style for beef cattle in particular -and the poor record for controlling other, more common cattle diseases, in that country. We also know precious little about FMDV survival in the environment (doi: 10.1016/j.coviro.2017.11.013) – an area that deserves as much attention as emergency vaccination. In other words, I am not as confident as the authors of this report seems to be – based on their conclusion – that emergency vaccination, whether with homologous or heterologous vaccine strains, are going to be the panacea for control of an FMD outbreak in Australia or elsewhere. It is recommended that the authors take this into account for their conclusions.

The manuscript could be further improved by also addressing the following issues:

Line 21: with reference to comments above it seems an overstatement to say that the cattle were “protected” – I recommend either elaborate on what they mean by protection or find an alternative term. Line 29: many readers – at least those not working directly with FMDV and cattle - may not know what a probang sample is, so better to say something like “oropharyngeal fluid (probang) sample”. Line 59: similarly, many readers would not know what an “r1 value”, so define it here – basically use the definition/explanation given in line 87 here at first mention. Section 2.2.: what was the gender of the calves? In multiple places later in the manuscript, the authors use the term “cow”, but that should really be heifer (or calf or animal), as the term ‘cow’ is in general in the English language literature defined as a female bovine greater than 3 years and/or having had 1-2 calves. Lines 135-141: I suggest providing this information in table format or a flow chart for clarity. Section 2.8: considering the importance of the RNA-data are in this report, it would be appropriate to provide more information about the test – not just a reference. See also later comment about actually presenting the data. Figures & Tables should be inserted as close to the text as possible, not at the end of the Results section. As per Instructions to Authors: “All Figures, Schemes and Tables should be inserted into the main text close to their first citation and must be numbered following their number of appearance” on the MDPI website. Lines 250, 257 and corresponding tables: what is the sensitivity of the virus isolation protocol relative to the qRT-PCR method used. Line 259 and elsewhere in the Results (e.g., lines 287-9) and in the Discussion, the authors keep mentioning “higher RNA levels” and “significant differences in the RNA levels”, but they are not presenting the data – all that is presented is a +/- (grey shading in the tables). The data should be presented – either in the body of the manuscript or in supplementary files. Similarly it would be appropriate for the authors to show the results of the mixed linear regression model and other analysis – at least in a supplementary file, as there is a reference to these types of analysis in several places, but the reader is left with no way of assessing it. Figures 1, 3: it is recommended to use standard deviation rather than standard error of the mean – see doi: 10.1124/jpet.119.264143. Discussion is in general not very clear and could benefit from a more succinct presentation of arguments. Lines 372-4: the authors rightly point out the lack of understanding of antibody-independent protection early in infection – so why not focus on that? Reference list: please correct the style of the references as per the instruction to authors at https://www.mdpi.com/journal/vaccines/instructions

Minor points:

Line 16: define FMDV at first usage. Line 18: reverse order of “21- and 7-day”. Line 129: change to “fetal bovine serum (FBS)” since that abbreviation is used later in the M&M section. Line 129: please provide information about the “antibiotics mix”. Line 221: replace “as well as” with “and” Line 245: replace “height” with “peak” Line 250: correct to “….. serum, but viral RNA …..” Line 252: correct to “…FMDV genome from serum on …..” Line 259: correct to “higher viral RNA levels …..” Figure legends should be placed under the figures, not above. Line 381: correct to “regression”

END

Author Response

In this manuscript Singanallur and colleagues describe their studies of efficacy of “emergency vaccination” against foot and mouth disease virus (FMDV) using heterologous virus strains with low in vitro antigenic match to the challenge virus. The authors employ a challenge model with intradermolingual inoculation of virus, which probably is a more severe challenge model than the natural infection route, but ensures that all animals receive the same amount of virus inoculum. They define clinical disease as lesions on the feet, while ignoring/disregarding any oral lesions. This seems odd since virus is more likely to be spread via saliva and nasal secretions and oropharyngeal fluid (probang samples) are used to assess viral persistence.

Response: We agree with the reviewer that virus is more likely to spread via oral and nasal excretions, but for vaccine efficacy studies such as ours, one needs to use challenge methods that will result in disease. However, this study focused more on protection against clinical disease.  For evaluation of FMD vaccines by challenge with infectious virus, the intradermolingual route of inoculation is both valid and recommended method by the Pharmacopoeias and the OIE Manual.  It is controlled, repeatable and reproducible.  Under this challenge method, the lesions on feet are considered as indication of generalisation and animals are classified as protected or not protected.  Both the potency (PD50) and the protection against podal generalisation (PPG) tests are based on this rule.

Challenge with FMDV was done 7 and 21 days post vaccination to simulate likely time frames during an epidemic, where vaccination rather than culling is a chosen manner of containment also combined with restrictions on animal movements. They saw protection against foot-lesions in all vaccinated animals except for two animals challenged 7 days post vaccination. However, virus was isolated from probang samples up to 30 days post challenge and viral RNA detected in all but one animal at 35 days post challenge (the last sampling day), indicating that vaccination does not protect against virus persistence and subclinical infection – as is also the experience from natural settings (e.g., DOI: 10.1111/tbed.12774; DOI: 10.1111/tbed.12963). Hence, even if movement restrictions are implemented at least until no further clinical signs are evident, there is no guarantee that the virus has been eradicated. While this may be of greater concern in endemic areas where probang testing of every animal might be beyond both practical and economic capabilities, it is worth taking into consideration for non-endemic areas, if FMDV is introduced and vaccination combined with movement restriction is considered an option. In high-density areas like Europe and parts of North-America, there would also have to be long-term control of fomites, trade etc as well as extensive testing. And in a country like Australia it would be next to impossible to test all cattle, considering the management style for beef cattle in particular -and the poor record for controlling other, more common cattle diseases, in that country. We also know precious little about FMDV survival in the environment (doi: 10.1016/j.coviro.2017.11.013) – an area that deserves as much attention as emergency vaccination. In other words, I am not as confident as the authors of this report seems to be – based on their conclusion – that emergency vaccination, whether with homologous or heterologous vaccine strains, are going to be the panacea for control of an FMD outbreak in Australia or elsewhere. It is recommended that the authors take this into account for their conclusions.

Response: We agree with the reviewer that vaccination will not be the ultimate to control FMD in Australia.  Controlling FMD in FMD free countries without vaccination is always a challenge.  There are many factors we need to consider for control and eventual freedom from disease, if an incursion occurs.  AusVet Plan describes in detail the response that Australia will take in case of an FMD incursion and all options are on table, including vaccination.  The decision to include vaccination as one of the many measures will be taken appropriately, based on scientific judgement and environmental factors.  This study identifies the vaccine strain that could be used in case of an incursion with A Asia/G-IX/SEA-97 strains, if it is decided to use vaccination as one of the tools for controlling the spread of the disease. 

To ensure this is clear, we’ve added the following to the end of the discussion:  ‘However, in free countries, vaccination should always be seen a one option that needs to be combined with other measures such as movement control and stamping out to ensure effective FMD eradication’.

The manuscript could be further improved by also addressing the following issues:

Line 21: with reference to comments above it seems an overstatement to say that the cattle were “protected” – I recommend either elaborate on what they mean by protection or find an alternative term.

Response: We have changed the abstract to read: At 7 dpv, three of five calves vaccinated with A MAY 97 and four of five vaccinated with A22 IRQ did not show lesions on the feet and were considered protected while at 21 dpv all five calves were protected with each vaccine, indicating equal efficacy of both vaccine strains.

Line 29: many readers – at least those not working directly with FMDV and cattle - may not know what a probang sample is, so better to say something like “oropharyngeal fluid (probang) sample”.

We have made the change as suggested

Line 59: similarly, many readers would not know what an “r1 value”, so define it here – basically use the definition/explanation given in line 87 here at first mention.

Changed to: The variant strains showed lower r1 values (poor antigenic match in an in vitro virus neutralization test or ELISA) when matched with the two vaccine strains, A22 Iraq 64 and A Malaysia 97, which are included in several (inter)national FMDV antigen banks (unpublished data). 

Section 2.2.: what was the gender of the calves? In multiple places later in the manuscript, the authors use the term “cow”, but that should really be heifer (or calf or animal), as the term ‘cow’ is in general in the English language literature defined as a female bovine greater than 3 years and/or having had 1-2 calves.

Changed calves throughout the manuscript

Lines 135-141: I suggest providing this information in table format or a flow chart for clarity.

Table 1 is furnished

Section 2.8: considering the importance of the RNA-data are in this report, it would be appropriate to provide more information about the test – not just a reference.

Details of the method included in section 2.8

See also later comment about actually presenting the data. Figures & Tables should be inserted as close to the text as possible, not at the end of the Results section. As per Instructions to Authors: “All Figures, Schemes and Tables should be inserted into the main text close to their first citation and must be numbered following their number of appearance” on the MDPI website.

Done as suggested

Lines 250, 257 and corresponding tables: what is the sensitivity of the virus isolation protocol relative to the qRT-PCR method used.

Response:  The virus isolation test used in this study is routinely used for laboratory detection of FMDV for diagnostic purposes and disease exclusion at the WVBR, Lelystad in the Netherlands.  These assays are well established, and documented assays and the results are quality controlled (ISO 17025 accredited).  When WVBR started using the realtime PCR, they did a comparison between virus isolation and PCR and found that the sensitivity of the real-time RT-PCR was comparable to the virus isolation test [98%(CI 93.9%-100%); Unpublished records of WVBR, Lelystad]. 

Line 259 and elsewhere in the Results (e.g., lines 287-9) and in the Discussion, the authors keep mentioning “higher RNA levels” and “significant differences in the RNA levels”, but they are not presenting the data – all that is presented is a +/- (grey shading in the tables). The data should be presented – either in the body of the manuscript or in supplementary files.

The data on RNA levels in the form of Cq values generated by the real-time RT-PCR is furnished in Supplementary Tables 1a and 1b.

Similarly, it would be appropriate for the authors to show the results of the mixed linear regression model and other analysis – at least in a supplementary file, as there is a reference to these types of analysis in several places, but the reader is left with no way of assessing it.

Provided as Supplementary data files 1, 2, 3 & 4.

Figures 1, 3: it is recommended to use standard deviation rather than standard error of the mean – see doi: 10.1124/jpet.119.264143.

We have changes this as suggested.

Discussion is in general not very clear and could benefit from a more succinct presentation of arguments.

We have addressed this concern and edited the discussion section

Lines 372-4: the authors rightly point out the lack of understanding of antibody-independent protection early in infection – so why not focus on that?

The was a discussion point and not the aim of the study.  No new data were presented to warrant further elaboration of this point.

Reference list: please correct the style of the references as per the instruction to authors at https://www.mdpi.com/journal/vaccines/instructions

Changed as suggested by using appropriate endnote style (MDPI.ens)

Minor points:

Line 16: define FMDV at first usage.

Addressed as per Reviewer 1 suggestion.

Line 18: reverse order of “21- and 7-day”.

Changed

Line 129: change to “fetal bovine serum (FBS)” since that abbreviation is used later in the M&M section.

Changed

Line 129: please provide information about the “antibiotics mix”.

Provided

Line 221: replace “as well as” with “and”

Changed

Line 245: replace “height” with “peak”

Changed

Line 250: correct to “….. serum, but viral RNA …..”

Changed

Line 252: correct to “…FMDV genome from serum on …..”

Changed

Line 259: correct to “higher viral RNA levels …..”

Changed

Figure legends should be placed under the figures, not above.

Changed

Line 381: correct to “regression”

Changed

Round 2

Reviewer 2 Report

This manuscript is much improved. The data is clearly presented and easy to understand.

Reviewer 3 Report

The authors should be commended for addressing this reviewer's comments and criticisms. The report may be more relevant for an Australian readership than the veterinary field at large, but the presentation is at least now clearer.